# TREE-STRUCTURED DECODING WITH DOUBLY-RECURRENT NEURAL NETWORKS

**David Alvarez-Melis & Tommi S. Jaakkola**
Computer Science and Artificial Intelligence Lab
MIT
{davidam,tommi}@csail.mit.edu

## ABSTRACT

We propose a neural network architecture for generating tree-structured objects from encoded representations. The core of the method is a *doubly* recurrent neural network model comprised of separate width and depth recurrences that are combined inside each cell (node) to generate an output. The topology of the tree is modeled explicitly together with the content. That is, in response to an encoded vector representation, co-evolving recurrences are used to realize the associated tree and the labels for the nodes in the tree. We test this architecture in an encoder-decoder framework, where we train a network to encode a sentence as a vector, and then generate a tree structure from it. The experimental results show the effectiveness of this architecture at recovering latent tree structure in sequences and at mapping sentences to simple functional programs.

## 1 INTRODUCTION

Recurrent neural networks have become extremely popular for modeling structured data. Key to their success is their ability to learn long-range temporal dependencies, their flexibility, and ease of customization. These architectures are naturally suited for modeling sequences since the underlying state evolution resulting from successive operations follows an inherently linear order (Williams & Zipser, 1995; Hochreiter & Schmidhuber, 1997). Indeed, they have been successfully adapted to language modeling (Zaremba et al., 2015), machine translation (Sutskever et al., 2014) and conversational agents (Vinyals & Le, 2015), among other applications.

Although sequences arise frequently in practice, other structures such as trees or graphs do not naturally conform to a linear ordering. For example, natural language sentences or associated parse trees, programs, hierarchical structures in biology, or molecules are not inherently linear structures. While sentences in natural language can be modeled as if they were linear sequences, the underlying process is compositional (Frege, 1892). Models that construct sentences compositionally should derive an advantage from adopting a more appropriate inductive bias.

The flexibility and success of recurrent neural networks in modeling and generating sequential data has prompted efforts to adapt them to non-sequential data too. Recent work has focused on the application of neural architectures to hierarchical structures, albeit in limited ways. Much of this work has assumed that either the full tree structure is given (Socher et al., 2012; Tai et al., 2015) or at least the nodes are (Socher & Lin, 2011; Chen & Manning, 2014; Kiperwasser & Goldberg, 2016). In the former scenario, the network aggregates the node information in a manner that is coherent with a given tree structure while, in the latter, generation is reduced to an attachment problem, i.e., sequentially deciding which pairs of nodes to join with an edge until a tree is formed.

The full problem of *decoding with structure*, i.e., generating a tree-structured object with node labels from a given vector representation, has remained largely unexplored until recently. Recent efforts to adapt RNNs to this context have so far remained relatively close to their sequential counterparts. For example, in order to capture depth and branching in the tree, one can introduce special tokens (Dong & Lapata, 2016) or use alternating RNNs coupled with external classifiers to predict branching (Zhang et al., 2016).

In this work, we propose a novel architecture tailored specifically to tree-structured decoding. At the heart of our approach is a *doubly-recurrent* (breadth and depth-wise recurrent) neural network which separately models the flow of information between parent and children nodes, and between siblings. Each of these relationships is modeled with a recurrent module whose hidden states are updated upon observing node labels. Every node in the tree receives two hidden states, which are then combined and used to predict a label for that node. Besides maintaining separate but simultaneous *fraternal* and *paternal* recurrences, the proposed architecture departs from previous methods in that it explicitly models tree topology. Each node in the network has modules that predict, based on the cell state, whether the node is terminal, both in terms of depth and width. Decoupling these decisions from the label prediction allows for a more concise formulation, which does not require artificial tokens to be added to the tree to simulate branching.

We test this novel architecture in various encoder-decoder frameworks, coupling it with sequential encoders to predict tree structure from encoded vector representations of sequences. The experimental results show the effectiveness of this approach at recovering latent structure in flattened string representations of trees (Section 4.1) and at mapping from natural language descriptions of simple programs to abstract syntax trees (Section 4.2). In addition, we show that even for sequence-to-sequence tasks such as machine translation, the proposed architecture exhibits desirable properties, such as invariance to structural changes and coarse-to-fine generation (Section 4.3).

To summarize, the main contributions of this paper are as follows:

- We propose a novel neural network architecture specifically tailored to tree-structured decoding, which maintains separate depth and width recurrent states and combines them to obtain hidden states for every node in the tree.

- We equip this novel architecture with a mechanism to predict tree topology *explicitly* (as opposed to *implicitly* by adding nodes with special tokens).

- We show experimentally that the proposed method is capable of recovering trees from encoded representations and that it outperforms state-of-the-art methods in a task consisting of mapping sentences to simple functional programs.

## 2  RELATED WORK

**Recursive Neural Networks.**    Recursive neural networks (Socher & Lin, 2011; Socher et al., 2012) were proposed to model data with hierarchical structures, such as parsed scenes and natural language sentences. Though they have been most successfully applied to encoding objects when their tree-structured representation is given (Socher et al., 2013), the original formulation by Socher & Lin (2011) also considered using them to *predict* the structure (edges), albeit for the case where nodes are given. Thus, besides their limited applicability due to their assumption of binary trees, recursive neural networks are not useful for *fully* generating trees from scratch.

**Tree-structured encoders.**    The Tree-LSTM of Tai et al. (2015) is a generalization of long short-term memory networks (Hochreiter & Schmidhuber, 1997) to tree-structured inputs. Their model constructs a sentence representation bottom-up, obtaining at every step the representation of a node in the tree from those of its children. In this sense, this model can be seen as a generalization of recursive neural networks to trees with degree potentially greater than two, with the additional long-range dependency modeling provided by LSTMs. They propose two methods for aggregating the states of the children, depending on the type of underlying tree: N-ary trees or trees with unknown and potentially unbounded branching factor. TreeLSTMs have shown promising results for compositional encoding of structured data, though by construction they cannot be used for decoding, since they operate on a given tree structure.

**Tree-structured decoders.**    Proposed only very recently, most tree-structured decoders rely on stacked on intertwined RNNs, and use heuristic methods for topological decisions during generation. Closest to our method is the Top-down Tree LSTM of Zhang et al. (2016), which generates a tree from an encoded representation. Their method relies on 4 independent LSTMs, which act *in alternation*—as opposed to simultaneously in our approach—yielding essentially a standard LSTM that changes the weights it uses based on the position of the current node. In addition, their method

provides children with *asymmetric parent input*: "younger" children receive information from the parent state only through the previous sibling's state. Though most of their experiments focus on the case where the nodes are given, they mention how to use their method for full prediction by introducing additional binary classifiers which predict which of the four LSTMs is to be used. These classifiers are trained in isolation after the main architecture has been trained. Contrary to this approach, our method can be trained end-to-end in only one pass, has a simpler formulation and explicitly incorporates topological prediction as part of the functioning of each neuron.

A similar approach is proposed by Dong & Lapata (2016). They propose SEQ2TREE, an encoder-decoder architecture that maps sentences to tree structures. For the decoder, they rely on hierarchical use of an LSTM, similar to Tai et al. (2015), but in the opposite direction: working top-down from the root of the tree. To decide when to change levels in the hierarchy, they augment the training trees with nonterminal nodes labeled with a special token $<n>$, which when generated during decoding trigger the branching out into a lower level in the tree. Similar to our method, they feed nodes with hidden representations of their parent and sibling, but they do so by concatenating both states and running them through a single recurrent unit, as opposed to our method, where these two sources of information are handled separately. A further difference is that our approach does not require artificial nodes with special tokens to be added to the tree, resulting in smaller trees.

**Hierarchical Neural Networks for Parsing.** Neural networks have also been recently introduced to the problem of natural language parsing (Chen & Manning, 2014; Kiperwasser & Goldberg, 2016). In this problem, the task is to predict a parse tree over a given sentence. For this, Kiperwasser & Goldberg (2016) use recurrent neural networks as a building block, and compose them recursively to obtain a tree-structured encoder. Starting from the leaves (words) they predict a parse tree with a projective bottom-up strategy, which sequentially updates the encoded vector representation of the tree and uses it to guide edge-attaching decisions. Though conceptually similar to our approach, their method relies on having access to the nodes of the tree (words) and only predicts its topology, so—similar to recursive neural networks—it cannot be used for a fully generative decoding.

## 3 DOUBLY RECURRENT NEURAL NETWORKS

Generating a tree-structured object from scratch using only an encoded representation poses several design challenges. First, one must decide in which order to generate the tree. If the nodes on the decoder side were given (such as in parsing), it would be possible to generate a tree bottom-up from these nodes (e.g. as Kiperwasser & Goldberg 2016 do). In the setting we are interested in, however, not even the nodes are known when decoding, so the natural choice is a top-down decoder, which starting from an encoded representation generates the root of the tree and then recursively generates the children (if any) of every node.

The second challenge arises from the asymmetric hierarchical nature of trees. Unlike the sequence-to-sequence setting where encoding and decoding can be achieved with analogous procedures, when dealing with tree-structured data these two involve significantly different operations. For example, an encoder that processes a tree bottom-up using information of a node's children to obtain its representation cannot be simply reversed and used as a decoder, since when generating the tree top-down, nodes have to be generated before their children are.

An additional design constraint comes from deciding what information to feed to each node. For sequences, the choice is obvious: a node should receive information from the node preceding or succeeding it (or both), i.e. there is a one-dimensional flow of information. In trees, there is an evident flow of information from parent to children (or vice-versa), but when generating nodes in a top-down order it seems unnatural to generate children in isolation: the label of one of them will likely influence what the states of the other children might be. For example, in the case of parse trees, generating a verb will reduce the chances of other verbs occurring in that branch.

With these considerations in mind, we propose an architecture tailored to tree decoding from scratch: top-down, recursive and *doubly-recurrent*, i.e. where both the *ancestral* (parent-to-children) and *fraternal* (sibling-to-sibling) flows of information are modeled with recurrent modules. Thus, the building block of a *doubly recurrent neural network* (DRNN) is a cell with two types of input states, one coming from its parent, updated and passed on to its descendants, and another one received from

its *previous* sibling,[1] updated and passed on to the next one. We model the flow of information in the two directions with separate recurrent modules.

Formally, let $\mathcal{T} = \{\mathcal{V}, \mathcal{E}, \mathcal{X}\}$ be a connected labeled tree, where $\mathcal{V}$ is the set of nodes, $\mathcal{E}$ the set of edges and $\mathcal{X}$ are node labels.[2] Let $g^a$ and $g^f$ be functions which apply one step of the two separate RNNs. For a node $i \in \mathcal{V}$ with parent $p(i)$ and previous sibling $s(i)$, the ancestral and fraternal hidden states are updated via

$$\mathbf{h}_i^a = g^a(\mathbf{h}_{p(i)}^a, \mathbf{x}_{p(i)}) \tag{1}$$

$$\mathbf{h}_i^f = g^f(\mathbf{h}_{s(i)}^f, \mathbf{x}_{s(i)}) \tag{2}$$

where $\mathbf{x}_{s(j)}, \mathbf{x}_{p(i)}$ are the vectors representing the previous sibling's and parent's values, respectively. Once the hidden depth and width states have been updated with these observed labels, they are combined to obtain a *predictive hidden state*:

$$\mathbf{h}_i^{(pred)} = \tanh\left(\mathbf{U}^f \mathbf{h}_i^f + \mathbf{U}^a \mathbf{h}_i^a\right) \tag{3}$$

where $\mathbf{U}^f \in \mathbb{R}^{n \times D_f}$ and $\mathbf{U}^a \in \mathbb{R}^{n \times D_a}$ are learnable parameters. This state contains combined information of the node's neighborhood in the tree, and is used to predict a label for it. In its simplest form, the network could compute the output of node $i$ by sampling from distribution

$$\mathbf{o}_i = \text{softmax}(\mathbf{W}\mathbf{h}_i^{(pred)}) \tag{4}$$

In the next section, we propose a slight modification to (4) whereby topological information is included in the computation of cell outputs. After the node's output symbol $\mathbf{x}_i$ has been obtained by sampling from $\mathbf{o}_i$, the cell passes $\mathbf{h}_i^a$ to all its children and $\mathbf{h}_i^f$ to the next sibling (if any), enabling them to apply Eqs (1) and (2) to realize their states. This procedure continues recursively, until termination conditions (explained in the next section) cause it to halt.

### 3.1 TOPOLOGICAL PREDICTION

As mentioned before, the central issue with free-form tree construction is to predict the topology of the tree. When constructing the tree top-down, for each node we need to decide: (i) whether it is a leaf node (and thus it should not produce offspring) and (ii) whether there should be additional siblings produced after it. Answering these two questions for every node allows us to construct a tree from scratch and eventual stop growing it.

Sequence decoders typically rely on special tokens to terminate generation (Sutskever et al., 2014). The token is added to the vocabulary and treated as a regular word. During training, the examples are padded with this token at the end of the sequence, and during testing, generation of this token signals termination. These ideas has been adopted by most tree decoders (Dong & Lapata, 2016). There are two important downsides of using a padding strategy for topology prediction in trees. First, the size of the tree can grow considerably. While in the sequence framework only one stopping token is needed, a tree with $n$ nodes might need up to $O(n)$ padding nodes to be added. This can have important effects in training speed. The second reason is that a single stopping token selected competitively with other tokens requires one to continually update the associated parameters in response to any changes in the distribution over ordinary tokens so as to maintain topological control.

Based on these observations, we propose an alternative approach to stopping, in which topological decisions are made *explicitly* (as opposed to implicitly, with stopping tokens). For this, we use the predictive hidden state of the node $\mathbf{h}^{(pred)}$ with a projection and sigmoid activation:

$$p_i^a = \sigma(\mathbf{u}^a \cdot \mathbf{h}_i^{(pred)}) \tag{5}$$

The value $p_i^a \in [0, 1]$ is interpreted as the probability that node $i$ has children. Analogously, we can obtain a probability of stopping *fraternal* branch growth after the current node as follows:

$$p_i^f = \sigma(\mathbf{u}^f \cdot \mathbf{h}_i^{(pred)}) \tag{6}$$

---

[1] Unlike the "ancestral" line, the order within sibling nodes is ambiguous. While in abstract trees it is assumed that the there is no such ordering, we assume that for the structures were are interested in learning there is always one: either chronological (the temporal order in which the nodes were generated) or latent (e.g. the grammatical order of the words in a parse tree with respect to their sentence representation).

[2] We assume throughout that these values are given as class indicators $\mathbf{x}_i \in \{1, \ldots, N\}$.

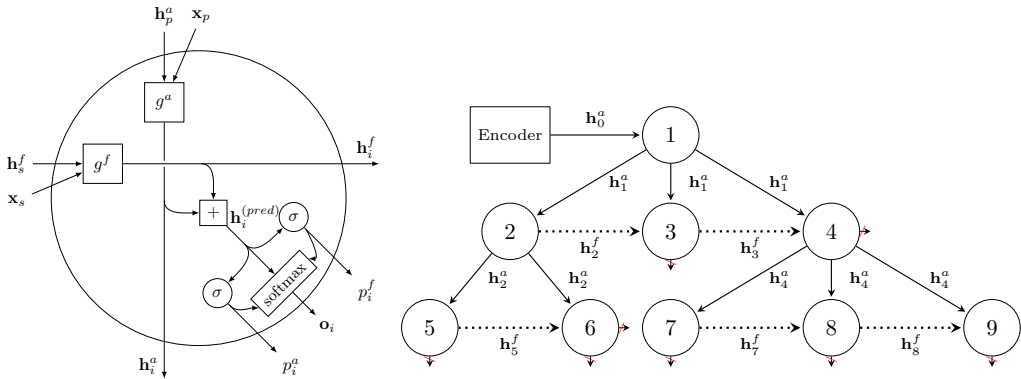

Figure 1: **Left**: A cell of the doubly-recurrent neural network corresponding to node $i$ with parent $p$ and sibling $s$. **Right**: Structure-unrolled DRNN network in an encoder-decoder setting. The nodes are labeled in the order in which they are generated. Solid (dashed) lines indicate ancestral (fraternal) connections. Crossed arrows indicate production halted by the topology modules.

Note that these stopping strategies depart from the usual padding methods in a fundamental property: the decision to stop is made *before* instead of *in conjunction* with the label prediction. The rationale behind this is that the label of a node will likely be influenced not only by its context, but also by the type of node (terminal or non-terminal) where it is to be assigned. This is the case in language, for example, where syntactic constraints restrict the type of words that can be found in terminal nodes. For this purpose, we include the topological information as inputs to the label prediction layer. Thus, (4) takes the form

$$\mathbf{o}_i = \text{softmax}(\mathbf{W}\mathbf{h}_i^{(pred)} + \alpha_i \mathbf{v}^a + \varphi_i \mathbf{v}^f) \qquad (7)$$

where $\alpha_i, \varphi_i \in \{0,1\}$ are binary variables indicating the topological decisions and $\mathbf{v}^a, \mathbf{v}^f$ are learnable offset parameters. During training, we use gold-truth values in (7), i.e. $\alpha_i = 1$ if node $i$ has children and $\varphi_i = 1$ if it has a succeeding sibling. During testing, these values are obtained from $p^a, p^f$ by sampling or beam-search. A schematic representation of the internal structure of a DRNN cell and the flow of information in a tree are shown in Figure 1.

### 3.2 TRAINING DRNNS

We train DRNNs with (reverse) back-propagation through structure (BPTS) (Goller & Kuechler, 1996). In the forward pass, node outputs are computed in a top-down fashion on the structure-unrolled version of the network, following the natural[3] dependencies of the tree. We obtain error signal at the node level from the two types of prediction: label and topology. For the former, we compute cross-entropy loss of $\mathbf{o}_i$ with respect to the true label of the node $\mathbf{x}_i$. For the topological values $p_i^a$ and $p_i^f$ we compute binary cross entropy loss with respect to gold topological indicators $\alpha_i, \varphi_i \in \{0,1\}$. In the backward pass, we proceed in the reverse (bottom-up) direction, feeding into every node the gradients received from child and sibling nodes and computing internally gradients with respect to both topology and label prediction. Further details on the backpropagation flow are provided in the Appendix.

Note that the way BPTS is computed implies and underlying decoupled loss function

$$\mathcal{L}(\widehat{\mathbf{x}}) = \sum_{i \in \mathcal{V}} \mathcal{L}^{label}(\mathbf{x}_i, \widehat{\mathbf{x}}_i) + \mathcal{L}^{topo}(\mathbf{p}_i, \widehat{\mathbf{p}}_i) \qquad (8)$$

The decoupled nature of this loss allows us to weigh these two objectives differently, to emphasize either topology or label prediction accuracy. Investigating the effect of this is left for future work.

---

[3]The traversal is always breadth-first starting from the root, but the order in which sibling nodes are visited might depend on the specific problem. If the nodes of the tree have an underlying order (such as in dependency parse trees), it is usually desirable to preserve this order.

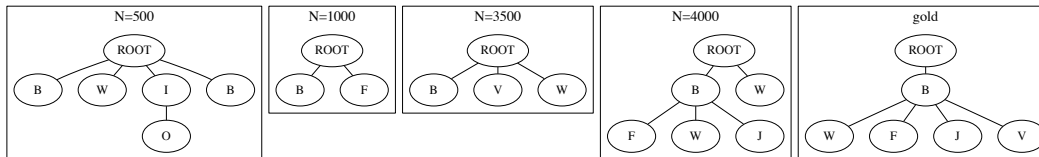

Figure 2: Trees generated by the DRNN decoder trained on subset of size $N$ of the synthetic dataset, for a test example with description "ROOT B W F J V".

As is common with sequence generation, during training we perform *teacher forcing*: after predicting the label of a node and its corresponding loss, we replace it with its gold value, so that children and siblings receive the correct label for that node. Analogously, we obtain the probabilities $p^a$ and $p^f$, compute their loss, and replace them for ground truth variables $\alpha_i, \varphi_i$ for all downstream computations. Addressing this *exposure bias* by mixing ground truth labels with model predictions during training (Venkatraman et al., 2015) or by incremental hybrid losses (Ranzato et al., 2016) is left as an avenue for future work.

## 4 EXPERIMENTS

### 4.1 SYNTHETIC TREE RECOVERY

In our first set of experiments we evaluate the effectiveness of the proposed architecture to recover trees from flattened string representations. For this, we first generate a toy dataset consisting of simple labeled trees. To isolate the effect of label content from topological prediction, we take a small vocabulary consisting of the 26 letters of the English alphabet. We generate trees in a top-down fashion, conditioning the label and topology of every node on the state of its ancestors and siblings. For simplicity, we use a Markovian assumption on these dependencies, modeling the probability of a node's label as depending only on the label of its parent and the last sibling generated before it (if any). Conditioned on these two inputs, we model the label of the node as coming from a multinomial distribution over the alphabet with a dirichlet prior. To generate the topology of the tree, we model the probability of a node having children and a next-sibling as depending only on its label and the depth of the tree. For each tree we generate a string representation by traversing it in breadth-first preorder, starting from the root. The labels of the nodes are concatenated into a string in the order in which they were visited, resulting in a string of $|\mathcal{T}|$ symbols. We create a dataset of 5,000 trees with this procedure, and split it randomly into train, validation and test sets (with a 80%,10%,10% split). Further details on the construction of this dataset are provided in the Appendix.

The task consists of learning a mapping from strings to trees, and using this learned mapping to recover the tree structure of the test set examples, given only their flattened representation. To do so, we use an encoder-decoder framework, where the strings are mapped to a fixed-size vector representation using a recurrent neural network. For the decoder, we use a DRNN with LSTM modules, which given the encoded representation generates a tree. We choose hyper-parameters with cross-validation. Full training details are provided in the Appendix.

Measuring performance only in terms of exact recovery would likely yield near-zero accuracies for most trees. Instead, we opt for a finer-grained metric of tree similarity that gives partial credit for correctly predicted subtrees. Treating tree generation as a retrieval problem, we evaluate the quality of the predicted tree in terms of the precision and recall of recovering nodes and edges present in the gold tree. Thus, we penalize both missing and superfluous components. As baseline, we induce a probabilistic context-free grammar (PCFG) on the full training data and use it to parse the test sentences. Note that unlike the DRNN, this parser has direct access to the sentence representation and thus its task is only to infer the tree structure on top of it, so this is indeed a strong baseline.

Figure 3 shows the results on the test set. Training on the full data yields node and edge retrieval F1-Scores of $75\%$ and $71\%$, respectively, the latter considerably above the baseline.[4] This $4\%$ gap can be explained by correct nodes being generated in the wrong part of the tree, as in the example in

---

[4]Since the PCFG parser has access to the nodes by construction, node accuracy for the baseline method is irrelevant and thus omitted from the analysis.

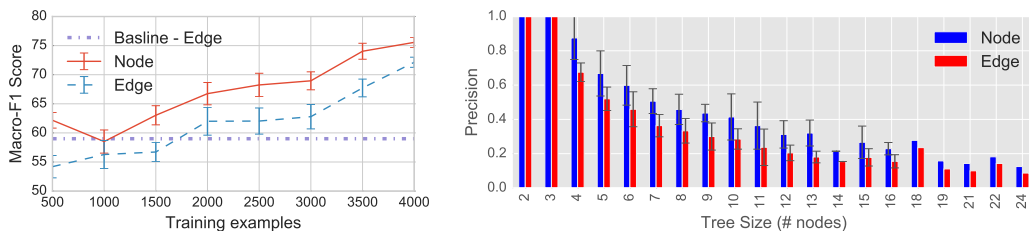

Figure 3: **Left**: F1-Score for models trained on randomly sampled subsets of varying size, averaged over 5 repetitions. **Right**: Node (first column) and edge (second) precision as a function of tree size.

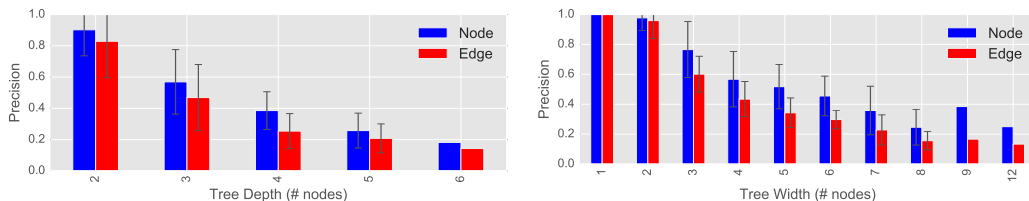

Figure 4: Node and edge precision as a function of tree depth (left figure) and width (right).

Figure 2. The second plot in Figure 3 shows that although small trees are recovered more accurately, precision decays slowly with tree size, with depth accounting for the largest effect (Figure 4).

## 4.2 MAPPING SENTENCES TO FUNCTIONAL PROGRAMS

Tree structures arise naturally in the context of programs. A typical compiler takes human-readable source code (expressed as sequences of characters) and transforms it into an executable abstract syntax tree (AST). Source code, however, is already semi-structured. Mapping natural language sentences directly into executable programs is an open problem, which has received considerable interest in the natural language processing community (Kate et al., 2005; Branavan et al., 2009).

The IFTTT dataset (Quirk et al., 2015) is a simple testbed for language-to-program mapping. It consists of if-this-then-that programs (called *recipes*) crawled from the IFTTT website[5], paired with natural language descriptions of their purpose. The recipes consist of a trigger and an action, each defined in terms of a channel (e.g. "*Facebook*"), a function (e.g. "*Post a status update*") and potentially arguments and parameters. An example of a recipe and its description are shown in Figure 5. The data is user-generated and extremely noisy, which makes the task significantly challenging.

---

[5] `www.ifttt.com`

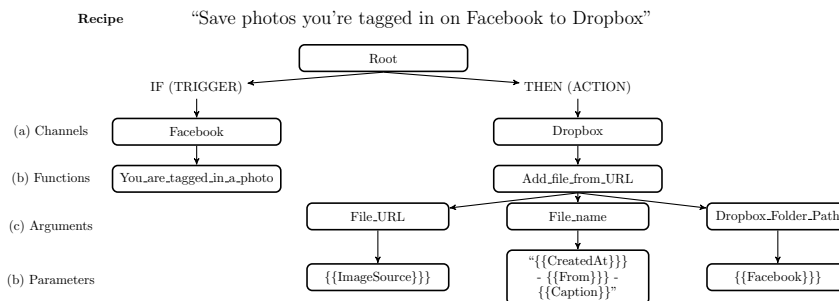

Figure 5: Example recipe from the IFTTT dataset. The description (above) is a user-generated natural language explanation of the if-this-then-that program (below).

Table 1: Results on the IFTTT task. **Left**: non-English and unintelligible examples removed (2,262 recipes). **Right**: examples for which at least 3+ humans agree with gold (758 recipes).

| Method | Channel | +Func | F1 | Method | Channel | +Func | F1 |
|---|---|---|---|---|---|---|---|
| retrieval | 36.8 | 25.4 | 49.0 | retrieval | 43.3 | 32.3 | 56.2 |
| phrasal | 27.8 | 16.4 | 39.9 | phrasal | 37.2 | 23.5 | 45.5 |
| sync | 26.7 | 15.4 | 37.6 | sync | 36.5 | 23.5 | 45.5 |
| classifier | 64.8 | 47.2 | 56.5 | classifier | 79.3 | 66.2 | 65.0 |
| posclass | 67.2 | 50.4 | 57.7 | posclass | 81.4 | 71.0 | 66.5 |
| Seq2Seq | 68.8 | 50.5 | 60.3 | Seq2Seq | 87.8 | 75.2 | 73.7 |
| Seq2Tree | 69.6 | 51.4 | 60.4 | Seq2Tree | 89.7 | **78.4** | 74.2 |
| Gru-Drnn | 70.1 | 51.2 | 62.7 | Gru-Drnn | 89.9 | 77.6 | 74.1 |
| Lstm-Drnn | **74.9** | **54.3** | **65.2** | Lstm-Drnn | **90.1** | 78.2 | **77.4** |

We approach this task using an encoder-decoder framework. We use a standard RNN encoder, either an LSTM or a GRU (Cho et al., 2014), to map the sentence to a vector representation, and we use a Drnn decoder to generate the AST representation of the recipe. We use the original data split, which consists of 77,495 training, 5,171 development and 4,294 test examples. For evaluation, we use the same metrics as Quirk et al. (2015), who note that computing exact accuracy on such a noisy dataset is problematic, and instead propose to evaluate the generated AST in terms of F1-score on the set of recovered productions. In addition, they compute accuracy at the channel level (i.e. when both channels are predicted correctly) and at the function level (both channels *and* both functions predicted correctly).

We compare our methods against the various extraction and phrased-based machine translation baselines of Quirk et al. (2015) and the the methods of Dong & Lapata (2016): Seq2Seq, a sequence-to-sequence model trained on flattened representations of the AST, and Seq2Tree, a token-driven hierarchical RNN. Following these two works, we report results on two noise-filtered subsets of the data: one with all non-English and unintelligible recipes removed and the other one with recipes for which at least three humans agreed with the gold AST. The results are shown in Table 1. In both subsets, Drnns perform on par or above previous approaches, with Lstm-Drnn achieving significantly better results. The improvement is particularly evident in terms of F1-score, which is the only metric used by previous approaches that measures global tree reconstruction accuracy. To better understand the quality of the predicted trees beyond the function level (i.e. (b) in Figure 5), we computed node accuracy on the arguments level. Our best performing model, Lstm-Drnn, achieves a Macro F1 score of 51% (0.71 precision, 0.40 recall) over argument nodes, which shows that the model is reasonably successful at predicting structure even beyond depth three. The best performing alternative model, Seq2Tree, achieves a corresponding F1 score of 46%.

## 4.3 Machine Translation

In our last set of experiments, we offer a qualitative evaluation DRNNs in the context of machine translation. Obtaining state-of-the-art results in machine translation requires highly-optimized architectures and large parallel corpora. This is not our goal. Instead, we investigate whether decoding with structure can bring benefits to a task traditionally approached as a sequence-to-sequence problem. For this reason, we consider a setting with limited data: a subset of the WMT14 dataset consisting of about 50K English $\leftrightarrow$ French sentence pairs (see the Appendix for details) along with dependency parses of the target (English) side.

We train a sequence-to-tree model using an LSTM encoder and a DRNN decoder as in the previous experiments. A slight modification here is that we distinguish left and right children in the tree, using two symmetric width-modules $g_L^f, g_R^f$ that produce children from the parent outwards. With this, children are lexically ordered, and therefore trees can be easily and un-ambiguously projected back into sentences. We compare our model against a sequence-to-sequence architecture of similar complexity (in terms of number of parameters) trained on the same data using the optimized Open-NMT library (Klein et al., 2017). For decoding, we use a simple best-of-k sampling scheme for our model, and beam search for the Seq2Seq models.

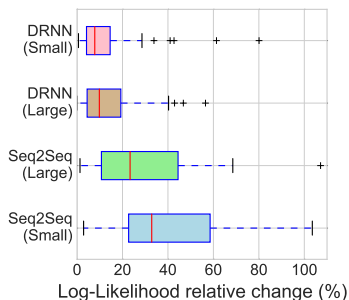

| Source | " produit différentes réponses qui changent avec le temps selon nos expériences et nos relations " | "je ne sais jamais quoi dire dans ces cas là" |
|---|---|---|
| SEQ2SEQ: | | |
| $l = 1$ | a | I |
| $l = 4$ | with the different actions | I do |
| $l = 8$ | with the different actions who change with | I do not know what to say |
| DRNN: | | |
| $d = 1$ | answers | know |
| $d = 2$ | different answers change | but i do not know |
| $d = 3$ | product the different answers change . | but i do not know to say |

Figure 6: Likelihood change under target structural perturbation.

Table 2: Translations at different resolutions (size constraints imposed during decoding) for two example sentences.

First, we analyze the quality of translations as a function of the maximum allowed target sentence "size". The notion of size for a sequence decoder is simply the *length* while for DRNN we use *depth* instead so as to tap into the inherent granularity at which sentences can be generated from this architecture. Two such examples are shown in Table 2. Since DRNN topology has been trained to mimic dependency parses top-down, the decoder tends to first generate the fundamental aspects of the sentence (verb, nouns), leaving less important refinements for deeper structures down in the tree. The sequence decoder, in contrast, is trained for left-to-right sequential generation, and thus produces less informative translations under max-length restrictions.

In our second experiment we investigate the decoders' ability to entertain natural paraphrases of sentences. If we keep the semantic content of a sentence fixed and only change its grammatical structure, it is desirable that the decoder would assign nearly the same likelihood to the new sentence. One way to assess this invariance is to compare the relative likelihood that the model assigns to the gold sentence in comparison to its paraphrase. To test this, we take 50 examples from the WMT test split and manually generate paraphrases with various types of structural alterations (see details in the Appendix). For each type of decoder, we measure the relative change (in absolute value) of the log-likelihood resulting from the perturbation. All the models we compare have similar standard deviation ($40 \pm 20$) of log-likelihood scores over these examples, so the relative changes in the log-likelihood remain directly comparable. For each architecture we train two versions of different sizes, where the sizes are balanced in terms of the number of parameters across the architectures. The results in Figure 6 show that DRNN's exhibit significantly lower log-likelihood change, suggesting that, as language models, they are more robust to natural structural variation than their SEQ2SEQ counterparts.

## 5  DISCUSSION AND FUTURE WORK

We have presented *doubly recurrent neural networks*, a natural extension of (sequential) recurrent architectures to tree-structured objects. This architecture models the information flow in a tree with two separate recurrent modules: one carrying ancestral information (received from parent and passed on to offspring) and the other carrying fraternal information (passed from sibling to sibling). The topology of the tree is modeled explicitly and separately from the label prediction, with modules that given the state of a node predict whether it has children and siblings.

The experimental results show that the proposed method is able to predict reasonable tree structures from encoded vector representations. Despite the simple structure of the IFTTT trees, the results on that task suggest a promising direction of using DRNNs for generating programs or executable queries from natural language. On the other hand, the results on the toy machine translation task show that even when used to generate sequences, DRNN's exhibit desirable properties, such as invariance over structural modifications and the ability to perform coarse-to-fine decoding. In order to truly use this architecture for machine translation, the approach must be scaled by resorting to batch processing in GPU. This is possible since forward and backward propagation are computed sequentially along tree traversal paths so that inputs and hidden states of parents and siblings can be grouped into tensors and operated in batch. We leave this as an avenue for future work.

ACKNOWLEDGEMENTS

DA-M acknowledges support from a CONACYT fellowship. The authors would like to thank the anonymous reviewers for their constructive comments.

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

## A   Variations on topology prediction

Besides the topology prediction approach presented in Section 3.1, we experimented with two additional variations of the proposed doubly-recurrent neuron: (i) using tokens to trigger both depth and width termination (i.e. implicit topology prediction) and (ii) using tokens for width-stopping decision, but predict explicitly depth termination (single topology prediction). Recall that in the model proposed in Section 3.1 both decisions are explicit (double topology prediction). The neurons in each of these alternative formulations are depicted in Figure 7. In order to train these two alternative models, we add special stopping tokens to the vocabulary, and we pad the training with additional nodes labeled with this token. Besides requiring larger trees and resulting in slower training, we empirically observed alternatives (i) and (ii) to result in worse performance. We hypothesize that this has to do with the fact that when using token-based stopping, topological and label prediction decisions are confounded, which results in less efficient learning.

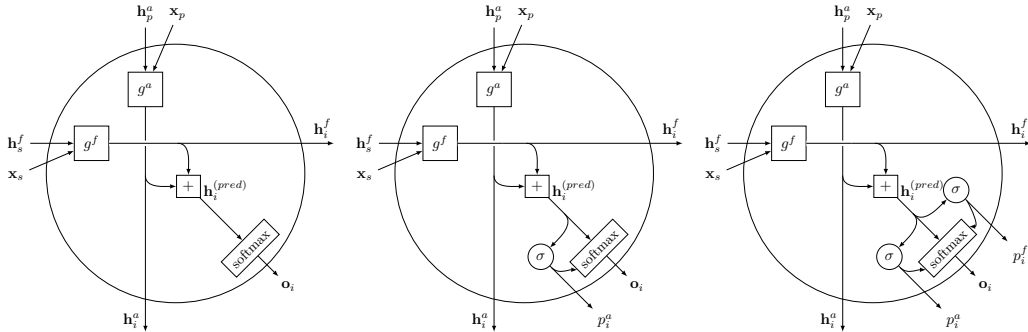

Figure 7: A single unit in each of the three alternative versions of the doubly-recurrent neural network, for node $i$ with parent $p$ and sibling $s$. **Left**: No explicit topology prediction, **Middle**: single (ancestral) topology prediction, **Right**: double (ancestral and fraternal) topology prediction. The top (left) incoming arrows represent the input and state received from the parent node (previous node, respectively).

## B   Training details

### B.1   Backpropagation with Drnn's

During training, we do the forward pass over the trees in breadth-first preorder, feeding into every node an ancestral and a fraternal state. For computational efficiency, before passing on the ancestral state to the offspring, we update it through the RNN using the current node's label, so as to avoid repeating this step for every child node. After the forward pass is complete, we compute label (cross-entropy) and topological (binary cross-entropy) loss for every node. In the backward pass, we compute in this order:

1. Gradient of the current node's label prediction loss with respect to softmax layer parameters $\mathbf{W}, \mathbf{v}^a, \mathbf{v}^f$: $\nabla_\theta \mathcal{L}(\mathbf{x}_i, \widehat{\mathbf{x}}_i)$.

2. Gradients of topological prediction variable loss with respect to sigmoid layer parameters: $\nabla_\theta \mathcal{L}(p_i^a, t_i^a)$ and $\nabla_\theta \mathcal{L}(p_i^f, t_i^f)$.

3. Gradient of predictive state layer parameters with respect to $\mathbf{h}^{(pred)}$.

4. Gradient of predicted ancestral and fraternal hidden states with respect to $g^f$ and $g^a$'s parameters.

The gradients of the input ancestral and fraternal hidden states are then passed on to the previous sibling and parent. When nodes have more than one child, we combine gradients from multiple children by averaging them. This procedure is repeated until the root note is reached, after which a single (ancestral state) gradient is passed to the encoder.

## B.2 Model Specification and Training Parameters

The best parameters for all tasks are chosen by performance on the validation sets. We perform early stopping based on the validation loss. For the IFTTT task, we initialize word embeddings with pretrained `GloVe` vectors (Pennington et al., 2014). For both tasks we clip gradients when the absolute value of any element exceeds 5. We regularize with a small penalty $\rho$ on the $l_2$ norm of the parameters. We train all methods with ADAM (Kingma & Ba, 2014), with initial learning rate chosen by cross-validation. The parameter configurations that yielded the best results and were used for the final models are shown in Table 3. Details about the four models used for the machine translation task are shown in Table 4.

Table 3: Hyperparameter choice for DRNNs in the synthetic and IFTTT tasks

| Task | Encoder | Dim | Batch | Learning Rate | Regularization $\rho$ |
|---|---|---|---|---|---|
| synthetic | LSTM | 50 | 20 | 0.05 | $1\times10^{-5}$ |
| IFTTT | GRU | 150 | 35 | 0.06 | $1\times10^{-4}$ |
| IFTTT | LSTM | 150 | 35 | 0.05 | $5\times10^{-4}$ |

Table 4: Models used in the machine translation task.

| Model | Encoder | Decoder | Dim | RNN Layers | Batch |
|---|---|---|---|---|---|
| Seq2Seq (Small) | LSTM | LSTM | 150 | 1 | 64 |
| Seq2Seq (Large) | LSTM | LSTM | 300 | 3 | 64 |
| Drnn (Small) | LSTM | DRNN-GRU (Left-Right) | 150 | 1 | 32 |
| Drnn (Large) | LSTM | DRNN-GRU (Left-Right) | 300 | 1 | 32 |

## C  Dataset Details

### C.1  Synthetic tree dataset generation

We generate trees in a top-down fashion, conditioning the label and topology of every node on the state of its ancestors and siblings. For simplicity, we use a Markovian assumption on these dependencies, modeling the probability of a node's label as depending only on the label of its parent $p(i)$ and the last sibling $s(i)$ generated before it (if any). Conditioned on these two inputs, we model the label of the node as coming from a multinomial distribution over the alphabet:

$$P(w_i \mid \mathcal{T}) = P(w \mid w_{p(i)}, w_{s(i)}) \sim \text{Multi}(\theta_{w_{p(i)}, w_{s(i)}}) \tag{9}$$

where $\theta_{w_{p(i)}, w_{s(i)}}$ are class probabilities drawn from a Dirichlet prior with parameter $\alpha_v$. On the other hand, we denote by $b_i^a$ the binary variable indicating whether node $i$ has descendants, and by $b_i^f$ that indicating whether it has an ensuing sibling. We model these variables as depending only on the label of the current node and its position in the tree:

$$P(b_i^a \mid \mathcal{T}) = P(b_i^a \mid w_i, D_i) = \text{Bernoulli}(p_{w_i}^a \cdot g^a(D_i))$$
$$P(b_i^f \mid \mathcal{T}) = P(b_i^f \mid w_i, W_i) = \text{Bernoulli}(p_{w_i}^f \cdot g^f(W_i))$$

where $D_i$ is the depth of node $i$ and $W_i$ its width, defined as its position among the children of its parent $p(i)$. Intuitively, we want to make $P(b_i^a = 1 \mid \mathcal{T})$ decrease as we go deeper and further along the branches of the tree, so as to control its growth. Thus, we model $g^a$ and $g^f$ as decreasing functions with geometric decay, namely $g^a(D) = (\gamma^a)^D$ and $g^f(W) = (\gamma^f)^W$, with $\gamma^a, \gamma^f \in (0, 1)$. For the label-conditioned branching probabilities $P(b_i^a \mid w_i)$ and $P(b_i^f \mid w_i)$, we use Bernoulli distributions with probabilities drawn from beta priors with parameters $(\alpha^a, \beta^a)$ and $(\alpha^f, \beta^f)$, respectively.

In summary, we use the following generative procedure to grow the trees:

1. For each $w_i \in V$, draw $p^a_{w_i} \sim \text{Beta}(\alpha^a, \beta^a)$ and $p^f_{w_i} \sim \text{Beta}(\alpha^f, \beta^f)$
2. For each pair $(w_i, w_j)$ draw $\theta_{w_i, w_j} \sim \text{Dir}(\alpha^V)$
3. While there is an unlabeled non-terminal node $i$ do:
   - Sample a label for $i$ from $w^* \sim P(w|w_{p(i)}, w_{s(i)}) = \text{Multi}(\theta_{w_{p(i)}, w_{s(i)}})$.
   - Draw $b_a \sim P(b^a|w^*, D) = \text{Bernoulli}(\gamma^D_a \cdot p^a_{w(i)})$, where $D$ is the current depth. If $b^a = 1$, generate an node $k$, set $p(k) = i$, and add it to the queue.
   - Draw $b_a \sim P(b^f|w^*, D) = \text{Bernoulli}(\gamma^W_f \cdot p^f_{w(i)})$, where $W$ is the current width. If $b^f = 1$, generate an node $k$, set $s(k) = i$, and add it to the queue.

Note that this generative process does create a dependence between the topology and content of the trees (since the variables $b^a$ and $b^f$ depend on the content of the tree via their dependence on the label of their corresponding node). However, the actual process by which labels and topological decision is generated relies on separate mechanisms. This is natural assumption which is reasonable to expect in practice.

The choice of prior parameters is done drawing inspiration from natural language parse trees. We want nodes to have low but diverse probabilities of generating children, so we seek a slow-decaying distribution with most mass allocated in values close to 0. For this, we use $(\alpha^a, \beta^a) = (0.25, 1)$. For sibling generation, we use $(\alpha^f, \beta^f) = (7, 2)$, which yields a distribution concentrated in values close to 1, so that nodes have on average a high and similar probability of producing siblings. Since we seek trees that are wider than they are deep, we use decay parameters $\gamma_a = 0.6, \gamma_f = 0.9$. Finally, we use a $\alpha_v = 10 \cdot \mathbf{1}$ for the parent-sibling probability prior, favoring non-uniform interactions. Using this configuration, we generate 5000 sentence-tree pairs, which we split into training (4000 examples), validation (500) and test (500) sets. The characteristics of the trees in the dataset are summarized in Table 5.

Table 5: Synthetic tree dataset statistics. Tree size is measured in number of nodes, depth is the largest path from the root node to a leaf and width is the maximum number of children for any node in the tree. The values reported correspond to means with one standard deviation in parentheses.

| Fold | Examples | Size | Depth | Width |
|------|----------|------|-------|-------|
| train | 4000 | 3.94 (3.38) | 1.42 (0.66) | 2.89 (1.71) |
| dev | 500 | 4.13 (3.21) | 1.46 (0.67) | 2.91 (1.76) |
| test | 500 | 3.64 (3.21) | 1.32 (0.61) | 2.80 (1.71) |

## C.2 IFTTT

The IFTTT dataset comes with a script to generate the data by crawling and parsing the recipes. Unfortunately, by the time we ran the script many recipes had been removed or changed. We therefore resorted to the original dataset used by Quirk et al. (2015). We converted these recipes into our tree format, assigning a node to each element in the first three levels (channels, functions and arguments, see figure 5). For the parameters level, many recipes have sentences instead of single tokens, so we broke these up creating one node per word. The last two layers are therefore the most topologically diverse, whereas the structure of the first two layers is constant (all trees have channels and functions). A very small fraction ($< 1\%$) of trees that could not by parsed into our format was excluded from the dataset.

Table 6 shows various statistics about the topological characteristics of the recipes in the IFTTT dataset. The middle columns show percentage of trees that contain nonempty arguments and parameters in trigger (IF) and action (THEN) branches. Almost all recipes have none empty arguments and parameters (and thus depth 4, excluding the root), and a lower percentage—but still a majority—has arguments and parameters on the trigger side too. The last two columns show tree statistics pertaining to the complexity of trees after conversion to our format. The distribution of tree sizes is mostly concentrated between 4 and 30 nodes, with a slow-decaying tail of examples above this range (see Figure 8).

Table 6: IFTTT dataset statistics. The middle columns show percentage of trees that contain nonempty arguments and parameters in trigger (IF) and action (THEN) branches. The last column shows average (with standard deviation) tree size and depth.

| Fold | Examples | Has args. (%) | | Has params. (%) | | Tree Size | |
|------|----------|---------|--------|---------|--------|----------|-----------|
| | | Trigger | Action | Trigger | Action | # Nodes | Depth |
| train | 67,444 | 69.10 | 98.46 | 65.47 | 96.77 | 16.93 (31.71) | 3.99 (.13) |
| dev | 4,038 | 69.44 | 98.46 | 66.42 | 96.31 | 16.55 (8.75) | 3.99 (.11) |
| test | 3,725 | 68.38 | 98.66 | 65.64 | 97.50 | 16.43 (8.18) | 3.99 (.12) |

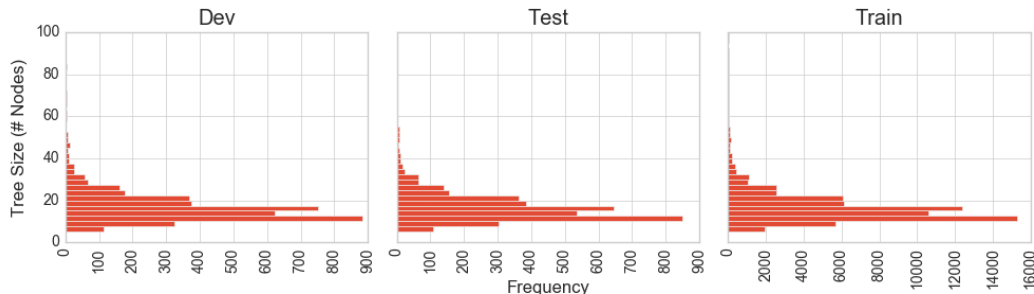

Figure 8: Tree size distribution in the IFTTT dataset.

Regarding the content of the trees, the labels of the nodes in the first two levels (channels and functions) come from somewhat reduced vocabularies: 111 and 434 unique symbols for the trigger branch, respectively, and 157 and 85 for the action branch. The lower layers of the tree have a much more diverse vocabulary, with about 60K unique tokens in total. On the source side, the vocabulary over the sentence descriptions is large too, with about 30K unique tokens. The average sentence size is 6.07 tokens, with 80% of the sentences having at most 12 tokens.

## C.3 MACHINE TRANSLATION

Starting from a preprocessed[6] 2% sub-selection of the English-French section of the WMT14 dataset, we further prune down the data by keeping only sentences of length between 5 and 20 words, and for which every word is within the 20K most frequent. The reason for this is to simplify the task by keeping only common words and avoiding out-of-vocabulary tokens. After this filtering, we are left with 53,607, 918 and 371 sentences for train, validation and test sets. After tokenizing, we obtain dependency parses for the target (English) sentences using the Stanford CoreNLP toolkit (Manning et al., 2014).

For the perturbation experiments, we randomly selected 50 sentences from among those in the test that could be easily restructured without significantly altering their meaning. The type of alterations we perform are: subordinate clause swapping, alternative construction substitution, passive/active voice change. In doing this, we try to keep the number of added/deleted words to a minimum, to minimize vocabulary-induced likelihood variations. When inserting new words, be verify that they are contained in the original vocabulary of 20K words. In Table 7 we show a few examples of the source, original target and perturbed target sentences.

---

[6]http://www-lium.univ-lemans.fr/ schwenk/cslm_joint_paper/

Table 7: Example structural perturbations for likelihood robustness experiments.

| | |
|---|---|
| source | "après un accord de paix signè en 1992 elle est devenue un parti d opposition." |
| target | "after a 1992 peace deal it became an opposition party." |
| perturbation | "it became an opposition party after a 1992 peace deal." |
| source | "cela représente environ 9 milliards de grains de maïs." |
| target | "that's about 9 billion individual kernels of corn." |
| perturbation | "this amounts to about 9 billion kernels of corn." |
| source | "l'exercice de fonctions publiques est une question de service public." |
| target | "public office is about public service." |
| perturbation | "the exercise of public functions is a matter of public service." |
| source | "nous avons ainsi effectué depuis la fin de l'hiver dernier 64 interventions." |
| target | "hence we have carried out 64 operations since last winter." |
| perturbation | "we have therefore carried out 64 operations since last winter." |
| source | "on estime qu'un enfant sur 2000 nés chaque année n'est ni un garcon ni une fille." |
| target | "an estimated one in 2000 children born each year is neither boy nor girl." |
| perturbation | "it is estimated that one in every 2000 children born every year is neither a boy nor a girl." |

# D    ADDITIONAL EXAMPLE GENERATED TREES

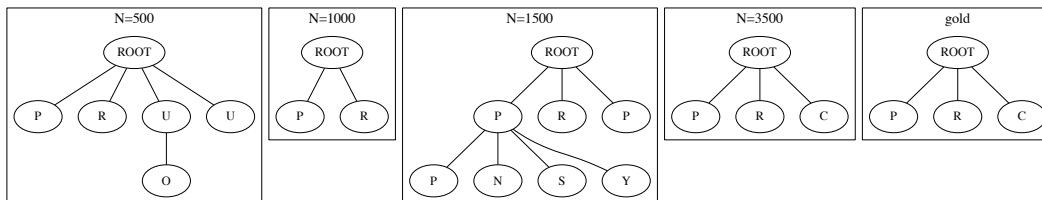

(a) Encoder sentence input: "ROOT P R C"

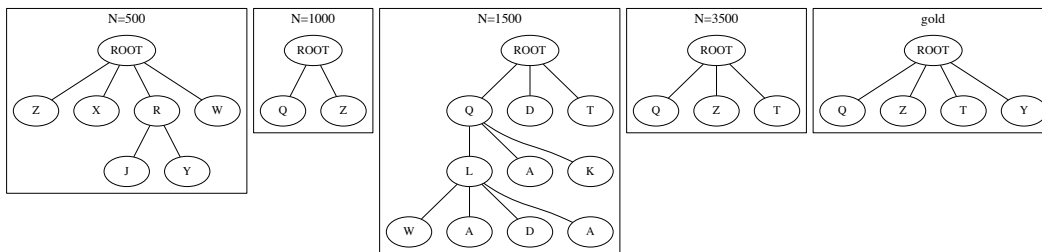

(b) Encoder sentence input: "ROOT Z T Y Q"

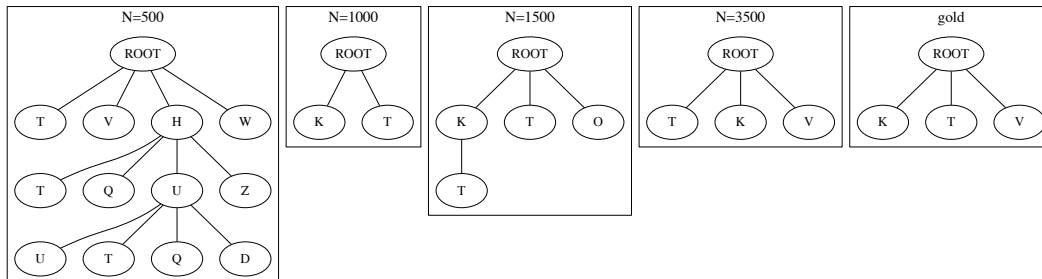

(c) Encoder sentence input: "ROOT K T V"

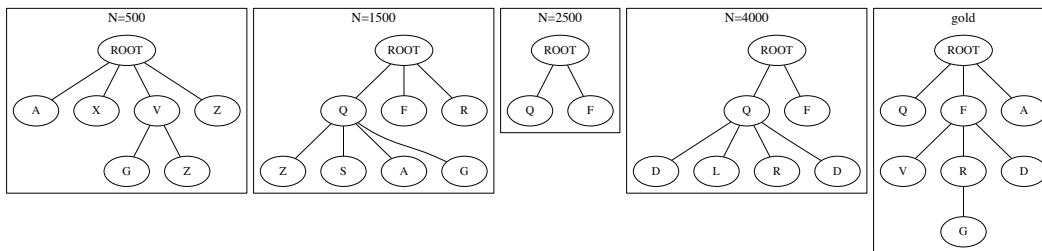

(d) Encoder sentence input: "ROOT Q F V R G D A"

Figure 9: Selected trees generated by the DRNN decoder from vector-encoded descriptions for test examples of the synthetic tree dataset. Trees in the same row correspond to predictions by models trained on randomly sampled subsets of size $N$ of the training split. We present cases for which the prediction is accurate (a,c) and cases for which it is not (b,d). Note how in (d) the model predicts many of the labels correctly, but confuses some of the dependencies (edges) in the tree.

