# Peer review of "Tree-structured decoding with doubly-recurrent neural networks"

_ICLR 2017 — accepted_

[Official Review · AnonReviewer4 · rating 7 · confidence 4 · 16 Dec 2016 (modified: 20 Jan 2017)]
**Accept**

Authors' response well answered my questions. Thanks. 
Evaluation not changed.

###

This paper proposes a neural model for generating tree structure output from scratch. The model does 1) separate the recurrence between depths and siblings; 2) separate the topology and label generation, and outperforms previous methods on a benchmark IFTTT dataset. Compared to previous tree-decoding methods, the model avoids manually annotating subtrees with special tokens, and thus is a very good alternative to such problems. The paper does solid experiments on one synthetic dataset, and outperforms alternative methods on one real-world IFTTT dataset. 

There are couple of interesting results in the paper that I believe is worth further investigation. Firstly, on the synthetic dataset, the precision drops rapidly with the number of nodes. Is it because that the vector representation of the sequential encoder fails to provide sufficient information of long sequences, such that the tree decoder can not do a good job? Or is it because that such tree decoder is not tolerant to the long sequence input, i.e., large tree structure? I believe that it is important to understand this before a better model can be developed. For example, if it is the fault of encoder, maybe an attention layer can be added, as in a seq-to-seq model, to preserve more information of the input sequence. 

Moreover, besides only showing how the precision changes with the number of nodes in the tree, it might be interesting to investigate how it goes with 1) number of depths; 2) number of widths; 3) symmetricity; etc. Moreover, as greedy search is used in decoding, it might be interesting to see how it helps, if it does, to use beam-search in tree decoding. 

On the IFTTT dataset, listing more statistics about this dataset might be helpful for better understanding the difficulty of this task. How deep are the trees? How large are the vocabularies on both language and program sides?

The paper is well written, except for minor typo as mentioned in my pre-review questions. 

In general, I believe this is a solid paper, and more can be explored in this direction. So I tend to accept it.

[Official Review · AnonReviewer2 · rating 6 · confidence 4 · 17 Dec 2016]

The paper propose DRNN as a neural decoder for tree structures. I like the model architecture since it has two clear improvements over traditional approaches — (1) the information flows in two directions, both from the parent and from siblings, which is desirable in tree structures (2) the model use a probability distribution to model the tree boundary (i.e. the last sibling or the leaf). This avoids the use of special ending symbols which is larger in size and putting more things to learn for the parameters (shared with other symbols).

The authors test the DRNN using the tasks of recovering the synthetic trees and recovering functional programs. The model did better than traditional methods like seq2seq models.

I think the recovering synthetic tree task is not very satisfying for two reasons — (1) the surface form itself already containing some of the topological information which makes the task easier than it should be (2) as we can see from figure 3, when the number of nodes grows (even to a number not very large), the performance of the model drops dramatically, I am not sure if a simple baseline only captures the topological information in the surface string would be much worse than this. And DRNN in this case, seems can’t show its full potentials since the length of the information flow in the model won’t be very long.

I think the experiments are interesting. But I think there are some other tasks which are more difficult and the tree structure information are more important in such tasks. For example, we have the seq2seq parsing model (Vinyals et al, 2014), is it possible to use the DRNN proposed here on the decoder side? I think tasks like this can show more potentials of the DRNN and can be very convincing that model architectures like this are better than traditional alternatives.

[Official Review · AnonReviewer3 · rating 6 · confidence 4 · 19 Dec 2016]
**No Title**

This paper proposes a variant of a recurrent neural network that has two orthogonal temporal dimensions that can be used as a decoder to generate tree structures (including the topology) in an encoder-decoder setting. The architecture is well motivated and I can see several applications (in addition to what's presented in the paper) that need to generate tree structures given an unstructured data.

One weakness of the paper is the limitation of experiments. IFTTT dataset seems to be an interesting appropriate application, and there is also a synthetic dataset, however it would be more interesting to see more natural language applications with syntactic tree structures. Still, I consider the experiments sufficient as a first step to showcase a novel architecture.

A strength is that the authors experiment with different design decisions when building the topology predictor components of the architecture, about when / how to decide to terminate, as opposed to making a single arbitrary choice.

I see future applications of this architecture and it seems to have interesting directions for future work so I suggest its acceptance as a conference contribution.

[Public Comment · (anonymous) · 22 Dec 2016]
**Some questions**

It is really a nice work and paper is written quite well. The related work section is comprehensive and the problem is well motivated. And in my view, the experiments are good enough especially the paper contribution is introducing a new model which can be very useful in generating structured outputs using recurrent structure.

Questions: 
q1) How long did it take to train each of the networks in the paper?
q2) Wondering any plan to release the code?


Thanks.

[Final Decision · Program Chairs · 06 Feb 2017]
**ICLR committee final decision**

The paper introduces a new model for generating trees decorated with node embeddings. Interestingly the authors do not assume that even leaf nodes in the tree are known a-priori. There has been very little work on this setting, and, the problem is quite important and general. Though the experiments are somewhat limited, reviewers generally believe that they are sufficient to show that the approach holds a promise.
 
 + an important and under-explored setting
 + novel model
 + well written
 
 - experimentation could be stronger (but seems sufficient -- both on real and artificial data)